# Influence of SARS-CoV-2 Status and Aging on the Nasal and Fecal Immunological Profiles of Elderly Individuals Living in Nursing Homes

**DOI:** 10.3390/v15061404

**Published:** 2023-06-20

**Authors:** Claudio Alba, Marta Mozota, Rebeca Arroyo, Natalia Gómez-Torres, Irma Castro, Juan Miguel Rodríguez

**Affiliations:** Department of Nutrition and Food Science, Complutense University of Madrid, 28040 Madrid, Spain; martamoz@ucm.es (M.M.); rebecaa@vet.ucm.es (R.A.); natgom07@ucm.es (N.G.-T.); irmacastro@ucm.es (I.C.)

**Keywords:** elderly, nursing home, SARS-CoV-2, COVID-19, feces, nasal samples, cytokines, chemokines

## Abstract

In the frame of SARS-CoV-2 infection, studies regarding cytokine profiling of mucosal-related samples are scarce despite being the primary infection sites. The objective of this study was to compare the nasal and fecal inflammatory profiles of elderly individuals living in a nursing home highly affected by COVID-19 (ELD1) with those of elderly individuals living in a nursing home with no cases of SARS-CoV-2 infection (ELD2) and, also, with those of healthy SARS-CoV-2-negative younger adults (YHA). BAFF/TNFSF13B, IL6, IL10 and TNF-α (immunological hallmarks of SARS-CoV-2 infection) were the only immune factors whose concentrations were different in the three groups. Their highest concentrations were achieved in the ELD1 group. Nasal and fecal concentrations of a wide number of pro-inflammatory cytokines were similar in the ELD1 and ELD2 groups but higher than those found in the YHA samples. These results reinforce the hypothesis that immunosenescence and inflammaging rendered the elderly as a highly vulnerable population to a neo-infection, such as COVID-19, which was evidenced during the first pandemic waves.

## 1. Introduction

Coronavirus disease 2019 (COVID-19), caused by severe acute respiratory syndrome coronavirus 2 (SARS-CoV-2), was recognized by the WHO as a pandemic on 11 March 2020. Recently (5 May 2023), the same organization declared the end of this disease as a global health emergency. During this 3-year period, this infectious disease was associated (and still is) with high mortality and morbidity rates worldwide, although the impact of the disease has been particularly strong among the elderly (≥65 years) [1]. This population was associated with the highest risks of hospitalization, severity, and mortality [2,3,4]. The high susceptibility of the elderly seems to be due to the confluence of most, if not all, COVID-19-related risk factors, including high rates of hypertension, cardiovascular disease, chronic respiratory diseases, diabetes, chronic kidney diseases, and malnutrition, among other comorbidities [5,6,7]. From the first waves of the pandemic to vaccine availability, elderly individuals living in nursing homes were the most affected group because of pre-COVID-19 conditions and practices and the accumulation of highly susceptible hosts in enclosed spaces, thus facilitating the spread of the virus in these facilities [6,8,9].

In addition, immunosenescence and so-called ”inflammaging” may play key roles in the enhanced severity of COVID-19 among the elderly [10]. Such processes involve aging-related changes in both innate and adaptive immunity [11,12], which are associated with an increase in all-cause mortality. Impairment in triggering effective adaptive immune responses together with a basal pro-inflammatory state [13] may notably decrease the ability of the elderly to properly control viral replication [14], predisposing them to more aggressive clinical consequences derived from cytokine storm and endothelial injury [10]. T cell responses are particularly affected, providing a rationale for the higher severity of COVID-19 among the elderly. T cells, and particularly Th1 cells, play a key role in the pathogenesis and outcomes of SARS-CoV-2 infection and are closely related to inflammation [15]. In fact, a prolonged Th1 cytokine profile has been observed in severely infected COVID-19 patients [16,17]. Most studies assessing the immune responses against SARS-CoV-2 have been focused on blood or bronchopulmonary samples; in contrast, studies investigating such responses in the other mucosal surfaces where the virus–host interactions start, such as that of the nasopharynx, are more scarce.

In this context, the objective of this study was to compare the nasal and fecal inflammatory profiles of elderly individuals living in a nursing home highly affected by COVID-19 with those of elderly individuals living in a nursing home with no cases of SARS-CoV-2 infection at the time of sampling and, also, with those of healthy SARS-CoV-2-negative younger adults (25–35 years old).

## 2. Materials and Methods

### 2.1. Study Participants

This study involved the recruitment of elderly individuals living at two nursing homes. The participants belonging to the first elderly group (ELD1; n = 22) were recruited at an elderly nursing home located in Moralzarzal (Madrid, Spain). This establishment had been strongly affected by the pandemic in the previous weeks before sampling (mortality rate: 38%). The participants of the second elderly group (ELD2; n = 29) were recruited at an elderly nursing home located in an area (Aínsa, Huesca, Spain) which had not been affected by the pandemic, yet. Finally, the third study group (HYA-: n = 20) consisted of healthy younger (25–35 years old) adults. Two samples (nasal wash and feces) were collected from each patient. The nasal wash was obtained using a standardized protocol [18]. Briefly, 1 mL of normal saline was instilled into one nostril, and the mix of saline and mucus was collected using an 8F suction catheter. Then, the process was repeated in the other nostril. After sampling from both nares, the catheter was washed by suctioning 2 mL of saline, and the washing saline was added to the samples to ensure that a standard volume of aspirate was obtained. All of the samples were immediately placed on ice and then stored at −80 °C until the immunological analyses were performed. The sampling collection period was extended from 15 April to 19 June 2020. Some demographic and health-related data (gender, age, and body mass index (BMI)) were also recorded at the sampling time. This work was conducted according to the guidelines laid down in the Declaration of Helsinki and was approved by the Ethics Committee of the Hospital Clínico San Carlos (Madrid, Spain) (protocol: CEIC 20/263-E_COVID; date of approval: 01/04/2020, act 4.1/20).

### 2.2. SARS-CoV-2 Detection

In order to know the current SARS-CoV-2 status of all of the participants at the sampling time, nasal samples were obtained using standard sterile swabs and submitted to SARS-CoV-2-specific reverse transcription polymerase chain reaction (RT-PCR) assays. Initially, RNA was extracted from the nasal swab samples by employing the Kingfisher Flex 96 extraction robot (Thermo Fisher Scientific, Waltham, MA, USA), the MagMax_Core_Flex extraction program, and the MagMAX Viral/Pathogen II Nucleic Acid Isolation kit (Thermo Fisher Scientific). Then, the TaqPath COVID-19 CE-IVD RT-PCR kit (Thermo Fisher Scientific) was used in a 384-well format with QuantStudio 7 Flex System equipment (Applied Biosystems, Waltham, MA, USA) for the detection of SARS-CoV-2. The three probes provided in this kit anneal to three SARS-CoV-2 target-specific genes encoding ORF1ab, N Protein, and S Protein. All of these assays were performed following the manufacturer’s instructions.

### 2.3. Immunoprofiling of the Nasal and Fecal Samples

The nasal samples (1 mL) were centrifuged, and the supernatants were used for the immunological assays. The fecal samples were prepared as described previously [19]. The concentrations of a wide array of inflammation-related immune factors (APRIL/TNFSF13, BAFF/TNFSF13B, Chitinase 3-like 1, IFNα, IFNβ, IFNγ, IL2, IL8, IL11, IL12p40, IL12p70, IL19, IL20, IL22, IL26, IL27p28, IL28/IFNλ2, IL29/IFNλ1, IL32, IL34, IL35, LIGHT/TNFSF14, MMP-1, MMP-2, MMP-3, osteocalcin, osteopontin, pentraxin 3, TSLP, TWEAK/TNFSF12, gp130/sIL-6Rb, sCD30/TNFRSF8, sCD163, sIL-6Ra, sTNF-R1, and sTNF-R2) were determined using a Bio-Plex Pro Human Inflammation Assay kit (Bio-Rad). In parallel, the nasal concentrations of IL6, IL10, and TNF-α were also measured using customized plates provided by Bio-Rad. All of the plates were read in the Bio-Plex 200 instrument (Bio-Rad, Hercules, CA, USA). Every assay was run in duplicate, and standard curves were performed for each analyte.

### 2.4. Statistical Analysis

Initially, the data distribution was assessed by implementing the Shapiro–Wilk normality test. Data found to be normally distributed were expressed as the mean alongside the 95% confidence interval (95% CI), while non-normally distributed data were represented as the median coupled with the interquartile range (IQR). To compare more than two groups of non-parametric data, we applied the Kruskal–Wallis test, followed by pairwise comparisons using the Wilcoxon rank-sum test with continuity correction for further exploration of specific group differences. For normally distributed data, we conducted an analysis of variance (ANOVA) for comparison across multiple groups, and if they were deemed significant, we followed them up with t-tests for pairwise comparisons. All statistical analyses were performed utilizing R-project software, version 4.0.3 (R-project, http://www.r-project.org; accessed on 12 March 2023), with a significance level set at *p* < 0.05 for all tests.

## 3. Results

### 3.1. Characteristics and SARS-CoV-2 Status of the Participants

The demographic and health-related data (age, gender, and body mass index) that were recorded at the sampling time are shown in Table 1. The mean age of the participants was approximately 85 years in both elderly groups and 30 years in the group of younger adults (Table 1). The three groups included the same or a very similar number of males and females. The mean BMI values ranged between 23 and 27, with the values being highest in the ELD2 group and lower in the ELD1 group, probably because of the nutritional consequences of COVID-19. A high percentage of the participants of the ELD1 group (n  =  18; ~81%) were SARS-CoV-2-positive; in contrast, all the participants of the groups ELD2 and YHA were negative at recruitment.

### 3.2. Assessment of the Immunological Parameters in the Nasal Samples

In relation to the nasal immunological profiles, APRIL/TNFSF13, BAFF/TNFSF13B, gp130/sIL-6Rb, osteopontin, pentraxin 3, IL6, and IL10 were the only immune factors that could be detected above their respective lower levels of quantification (LLOQ) in all of the samples from the three study groups (Table 2 and Appendix A). The frequency of detection was high (≥75% of the samples of the three groups) for the following immune factors: chitinase 3-like 1, IL8, IL32, and TWEAK/TNFSF12 (Table 2 and Appendix A). In the cases of TNF-α, sTNF.R1, sTNF.R2, TSLP, and LIGHT/TNFSF14, their frequencies of detection were high in the ELD1 and ELD2 groups but much lower (or not detected) in the YHA group, while the opposite was observed for IL19. The rest of the immune factors tested in this study (IFNα, IFNβ, IFNγ, IL2, IL11, IL12p40, IL12p70, IL20, IL22, IL26, IL27p28, IL28/IFNλ2, IL29/IFNλ1, IL34, IL35, MMP-1, MMP-2, MMP-3, osteocalcin, sCD30/TNFRSF8, sCD163, and sIL-6Ra) could be detected above their LLOQs in none or in a very small percentage of the samples (Appendix A).

In relation to the concentrations of those immune factors that were detected in the nasal samples, no statistically significant differences were found for chitinase 3-like 1, gp130/sIL-6Rb, and TWEAK/TNFSF12 when the three groups were compared. The only immune factors whose concentrations were significantly different in the three groups were BAFF/TNFSF13B, IL6, IL10, and TNF-α. The concentrations of these four compounds in the nasal samples were higher in the ELD1 group than in the ELD2 group and higher in the ELD2 group than in the YHA group (Table 2).

In the cases of IL8, IL32, LIGHT/TNFSF14, sTNF.R1, sTNF.R2, and TSLP, their concentrations were significantly higher in the ELD1 and ELD2 samples than in those of the YHA group (Table 2). Although the values for most of these immune factors tended to be higher in the ELD1 samples than in the ELD2 ones, the differences did not reach statistical significance (Table 2). The contrary was observed for IL19 and pentraxin 3 since the median values of these two immune factors were higher in the YHA group than in the other two groups (Table 2). Finally, the concentrations of APRIL/TNFSF13 and osteopontin in the ELD1 samples were higher than in the YHA samples, while the ELD2 samples were statistically similar to either ELD1 or YHA samples in relation to these two immune factors (Table 2).

### 3.3. Assessment of the Immunological Parameters in the Fecal Samples

BAFF/TNFSF13B, osteopontin, pentraxin 3, and TWEAK/TNFSF12 were the only immune factors that could be detected in all the fecal samples from the three study groups (Table 3 and Appendix A). The frequency of detection was very high (≥90% of the samples of the three groups) for chitinase 3-like 1, gp130/sIL-6Rb, and IL32, and high (≥70% of the samples of the three groups) for IL34, IL35, and TSLP (Table 3 and Appendix A).

The percentage of samples in which APRIL/TNFSF13, IL8, IL26, and LIGHT/TNFSF14 were detected was lower (30–60%) but similar among the three groups. In the case of sTNF.R1 and sTNF.R2, their frequencies of detection were very high (100%) in the ELD1 and ELD2 groups but somehow lower (≤70%) in the YHA group, while the opposite was observed for pentraxin 3 (100% in the YHA group versus 30% in the ELD1 and ELD2 groups) and IL19 (70% in the YHA groups versus 50–60% in the ELD1 and ELD2 groups). The rest of the immune factors tested in this study (IFNα, IFNβ, IFNγ, IL2, IL11, IL12p40, IL12p70, IL20, IL22, IL27p28, IL28/IFNλ2, IL29/IFNλ1, MMP-1, MMP-2, MMP-3, osteocalcin, sCD30/TNFRSF8, sCD163, and sIL-6Ra) could be detected above their LLOQs in a very small percentage of samples (Appendix A).

In relation to the concentrations of those immune factors that were detected in the fecal samples, no statistically significant differences were found for APRIL/TNFSF13, chitinase 3-like 1, gp130/sIL-6Rb, IL8, IL26, IL34, IL35, LIGHT/TNFSF14, osteopontin, and TWEAK/TNFSF12 when the three groups were compared (Table 3). None of the immune factors quantified in this work displayed concentrations that were significantly different in the three groups.

In the cases of BAFF/TNFSF13B, IL32, sTNF.R1, sTNF.R2, and TSLP, their concentrations were significantly higher in the ELD1 and ELD2 samples than in those of the YHA group (Table 3). Although the BAFF/TNFSF13B values tended to be higher in the ELD1 samples than in the ELD2 ones, the differences did not reach statistical significance (Table 3). The contrary was observed for IL19 and pentraxin 3 since the median values of these two immune factors were higher in the YHA group than in the other two groups (Table 3).

When the nasal and fecal samples were compared, some differences were found. All three groups (ELD1, ELD2, and YHA) were significantly different from each other in relation to the nasal concentration of BAFF.TNFSF13B; in contrast, no differences were found in the fecal samples between the two elderly groups. In addition, the concentrations of some factors (APRIL.TNFSF13, LIGHT.TNFSF14, and osteopontin) were shown to be statistically similar in the fecal samples of the three groups, while their nasal concentrations were higher in the samples of the ELD1 group than in those from the YHA group.

## 4. Discussion

Up to the present, most immunological studies focused on COVID-19 have analyzed the presence of anti-SARS-CoV-2 antibodies and several chemokines and cytokines (mainly those involved in the so-called cytokine storm) in blood samples. However, studies regarding immunoprofiling of mucosal-related samples are scarce despite the discordances observed between the serum and nasal cytokine concentrations during viral respiratory infections [20,21], including that caused by SARS-CoV-2 [22]. As a result, the key roles played by the mucosal immune system, particularly in the early infection stages, are underestimated when cytokine assessments of SARS-CoV-2 immunopathology are focused exclusively on the blood compartment [22]. Serum cytokines do not fully reflect viral responses at mucosal surfaces, including those existing in primary infection sites, such as the upper respiratory tract. Dissemination of SARS-CoV-2 to the lower respiratory tract may be avoided or minimized when its replication is efficiently controlled in the upper respiratory tract, leading to either an asymptomatic or a milder infection. In addition, the gut mucosal surface may also be relevant because of its role as a potent effector of immune responses within the mucosal-associated lymphoid tissue. In fact, it has been shown that the gut is a relevant site of innate and adaptive immune responses to SARS-CoV-2 and that gut dysbiosis and inflammation may contribute to systemic inflammation and may be associated with severe COVID consequences [23]. In this context, we compared the nasal and fecal inflammatory profiles of elderly individuals living in two nursing homes, one of them highly affected by COVID-19 and the other one with no cases of SARS-CoV-2 infection at the time of sampling. In addition, we also analyzed the nasal and fecal inflammatory profiles of a group of healthy SARS-CoV-2-negative younger adults. The idea of comparing these three groups was to elucidate the potential influence of SARS-CoV-2 status and age in the nasal and fecal immunological patterns.

BAFF/TNFSF13B was one of the four immune factors analyzed in this study in which nasal concentrations were significantly different in the three groups (from the highest to the lowest: ELD1, ELD2, and YHA). Production of BAFF sharply increases in the respiratory tract during inflammatory and infectious processes, including infections by the respiratory syncytial virus (RSV) and cystic fibrosis. As a result, its concentration in the nasal and bronchoalveolar lavage fluids collected from such patients is often very high [24,25,26]. Recently, high nasal concentrations have also been reported in COVID-19-suffering elderly patients [27]. BAFF/TNFSF13B and APRIL/TNFSF13 are two members of the TNF superfamily that play key roles in the activation and differentiation of B cells. Analysis of the expression of the genes encoding these two cytokines (TNFSF13B and TNFSF13, respectively) by cells obtained from samples of bronchoalveolar lavage fluid of COVID-19 patients has shown that TNFSF13B is highly expressed by both macrophages and neutrophils, while TNFSF13 was expressed only by macrophages [28]. The same study revealed that macrophages from patients with severe COVID-19 disease had high expression of TNFSF13B but much lower expression of TNFSF13 [24]. This may explain why only the BAFF/TNFSF13B concentration was different between the nasal ELD1 and nasal ELD2 samples. A significant decrease in the nasal concentrations of BAFF/TNFSF13B was the most relevant immunological hallmark associated with the benefits of the administration of a probiotic strain to SARS-CoV-2- infected elderly individuals living in a nursing home [23].

IL6, IL10, and TNF-α were the other three immune factors whose nasal concentrations were significantly different in the three groups (and, again, highest in ELD1, middle in ELD2, and lowest in YHA). These three immune factors are considered among the hallmark cytokines associated with SARSCov2 infection and high concentrations can be detected in nasal samples of COVID-19 patients [18,25,26]. The entry of SARS-CoV-2 into target respiratory cells induces the expression of Th1 cells and activates hyperinflammatory signal transducers, leading to enhanced levels of IL6, TNF-α, and IL10, accompanied by other cytokines and by infiltration of macrophages and neutrophils [17,21,29]. The nasal mucosal surface is a very active site of lasting cytokine responses against SARS-CoV-2 and, in fact, high IL-6 levels in the nasal environment may remain for some weeks after symptom onset [21].

Therefore, it is not strange that the ELD1 group, characterized by a high percentage of SARS-CoV-2-infected individuals, was the one with the highest nasal concentrations of BAFF/TNFSF13B, IL6, IL10, and TNF-α. It is also not surprising that their concentrations were higher in group ELD2 than in group YHA. Both groups only included SARS-CoV-2-negative participants, but the first was composed of elderly individuals, a population that includes a percentage of people suffering from chronic respiratory diseases, while the second was integrated with healthy younger adults. Increases in such immune factors have also been described in other infectious and inflammatory conditions affecting the respiratory tract [23,24,25,26,30,31].

Interestingly, the nasal concentrations of a wide number of the cytokines analyzed in this work (IL8, IL19, IL32, LIGHT/TNFSF14, pentraxin 3sTNF.R1, sTNF.R2, and TSLP) were statistically similar in the ELD1 and ELD2 groups but significantly higher (IL8, IL32, LIGHT/TNFSF14, sTNF.R1, sTNF.R2, and TSLP) or lower (IL19 and pentraxin 3) than those found in the YHA samples. The results were similar in relation to the fecal samples, where many of the immune factors that were detected above their respective LLOQs (BAFF/TNFSF13B, IL19, IL32, pentraxin 3, sTNF.R1, sTNF.R2, and TSLP) were statistically similar in the ELD1 and ELD2 groups but significantly higher (BAFF/TNFSF13B, IL32, sTNF.R1, sTNF.R2, and TSLP) or lower (IL19 and pentraxin 3) than those found in the YHA samples. These results show that, while SARS-CoV-2 status seems to be driven by the differences in the nasal concentrations of BAFF/TNFSF13B, IL6, IL10, and TNF-α between the ELD1 and the ELD2 groups, the age factor drives the differences between, on the one hand, the ELD1 and ELD2 groups and, on the other hand, the YHA group. The fact that differences were found when some parameters were measured in different biological samples (nasal or fecal samples) provided by the same subject or by the same study group indicates that some cytokines and chemokines may have different patterns in different mucosal surfaces, even within a same host, and, therefore, confirms that the type of sample has to be taken into account when interpreting immunological data. 

It has been repeatedly reported that aging is one of the most prominent risk factors for the outcomes of SARS-CoV-2 infection, with higher rates of COVID-19-related morbidity, complications, hospitalizations, ICU admissions, and death among elderly individuals (>65 years) than among any age group [2,32]. Immunosenescence and inflammaging are age-related processes that involve changes in both innate and adaptive immunity and a low-grade inflammatory status, leading to decline and dysregulation of immune function in older adults, which increases all-cause mortality [10,33]. Overall, and in contrast to infants and younger adults, older adults are less capable of responding to neo-antigens, such as SARS-CoV-2 because of the lower rate of naïve T cells, uncontrolled activation of innate responses, the inability to drive effective adaptive immune responses, and a higher pro-inflammatory state at both the systemic and mucosal levels [10,32]. Interestingly, BAFF/TNFSF13B is considered as a potential anti-aging target since its concentration is particularly elevated in senescent cells and suppression of its production reduces IL6 secretion and senescent-associated phenotypes [34]. In fact, suppression of this cytokine has been suggested as a therapeutic approach to minimize aging-dependent insulin resistance [35].

## 5. Conclusions

Nasal concentrations of BAFF/TNFSF13B, IL6, IL10, and TNF-α were significantly higher in samples from elderly individuals than in those from younger healthy adults. Regarding the elderly individuals, they were higher among elderly individuals living in a nursing home with a high rate of SARS-CoV-2-infected subjects, indicating that they can be considered as immunological hallmarks of SARS-CoV-2 infection at the nasopharyngeal level. Nasal and fecal concentrations of a wide number of pro-inflammatory cytokines were similar in the ELD1 and ELD2 groups but higher than those found in the YHA samples. These results reinforce the hypothesis that immunosenescence and inflammaging rendered the elderly as a highly vulnerable population to a neo-infection, such as COVID-19, which was evidenced during the first pandemic waves.

In this context, the increased nasal and fecal concentrations of most of the pro-inflammatory cytokines analyzed in this study in the ELD2 group, in relation to the YHA group, predicts that if the participants of these two groups were infected by SARS-CoV-2, the consequences for those in the ELD2 group would be much worse than among the YHA population. In order to confirm or extend the findings reported in this work, nasal and fecal samples should be collected periodically from participants in longitudinal life studies so that the adult life course trajectories of the immune factors can be studied and parameterized mathematically.

## Figures and Tables

**Table 1 viruses-15-01404-t001:** Main characteristics of the participants in this work.

	ELD1 (n = 22)	ELD2 (n = 29)	YHA (n = 20)
Age (years old)	84.9 (81.6–88.2)	84.6 (81.9–87.3)	29.5 (28.2–30.7)
Gender			
Male	11 (50%)	14 (48%)	10 (50%)
Female	11 (50%)	15 (52%)	11 (50%)
BMI	24.8 (23.0–26.6)	27.0 (25.7–28.4)	23.8 (23.3–24.4)
SARS-CoV-2-positive	18 (80.95%)	0 (0%)	0 (0%)

**Table 2 viruses-15-01404-t002:** Frequencies of detection and concentrations (mean (IQR)) of the selected immune factors in the nasal samples of the three study groups (YHA, ELD1, and ELD2) #. All of the concentrations are expressed as ng/L, with the exception of BAFF/TNFSF13B, APRIL/TNFSF13, and chitinase 3-like 1, which are expressed as μg/L.

	YHA		ELD2		ELD1	
	n (%)	Concentration	n (%)	Concentration	n (%)	Concentration
APRIL/TNFSF13	20 (100%)	3878.76 [2499.49–4405.33]**a**#	29 (100%)	4948.15 [2023.01–8214.25]**ab**	22 (100%)	7353.45 [3520.253–9469.71]**b**
BAFF/TNFSF13B	20 (100%)	470.705 [338.91–499.39]**a**	29 (100%)	1775.64 [1347.92–2907.64]**b**	22 (100%)	2695.09 [1973.85–3409.62]**c**
Chitinase 3-like 1	18 (90%)	808.32 [444.50–1082.39]a	24 (82.76%)	895.01 [567.32–1089.25]a	17 (77.27%)	874.65 [286.35–1088.28]a
gp130/sIL-6Rb	20 (100%)	142.23 [106.92–234.24]a	29 (100%)	160.03 [121.30–202.91]a	22 (100%)	137.39 [102.71–200.68]a
IL8	16 (80%)	0.32 [0.17–0.67]**a**	25 (86.21%)	2.09 [1.15–2.43]**b**	18 (81.82%)	2.50 [1.44–2.96]**b**
IL6	20 (100%)	0.37 [0.31–0.44]**a**	29 (100%)	0.83 [0.56–1.17]**b**	22 (100%)	1.25 [1.16–1.40]**c**
IL10	20 (100%)	0.425 [0.39–0.4625]**a**	29 (100%)	1.42 [0.89–2.51]**b**	22 (100%)	5.24 [3.00–10.91]**c**
IL19	15 (75%)	4.30 [1.90–6.45]**a**	10 (34.48%)	3.64 [0.69–2.67]**b**	8 (36.36%)	2.73 [0.64–0.9625]**b**
IL32	15 (75%)	1.78 [0.88–3.18]**a**	29 (100%)	5.01 [3.96–7.02]**b**	18 (81.82%)	6.33 [3.30–7.20]**b**
LIGHT/TNFSF14	5 (25%)	0.62 [0.07–1.08]**a**	15 (51.72%)	1.16 [0.23–1.21]**ab**	13 (59.09%)	1.18 [0.22–1.31]**b**
Osteopontin	20 (100%)	61.25 [51.78–71.03]**a**	29 (100%)	77.22 [49–98.47]**ab**	22 (100%)	85.095 [69.05–99.815]**b**
Pentraxin-3	20 (100%)	3.84 [2.97–4.7875]**a**	11 (37.93%)	1.90 [0.64–0.99]**b**	7 (31.82%)	1.64 [0.39–0.525]**b**
sTNF-R1	12 (60%)	12.42 [2.14–21.15]**a**	29 (100%)	19.25 [16.87–34.03]**b**	21 (95.45%)	27.80 [18.20–40.23]**b**
sTNF-R2	6 (30%)	4.51 [0.14–7.46]**a**	29 (100%)	10.74 [7.96–17.21]**b**	21 (95.45%)	11.56 [7.44–18.50]**b**
TNF-α (pg/mL)	0 (0%)	Nd (LLOQ: <0.709)**a**	22 (75.86%)	0.88 [0.709–1.24]**b**	22 (100%)	1.38 [1.19–1.875]**c**
TSLP	3 (15%)	0.21 [0.16–>0.31]**a**	25 (86.21%)	0.66 [0.49–0.80]**b**	17 (77.27%)	0.56 [0.39–0.76]**b**
TWEAK/TNFSF12	17 (85%)	2.00 [0.71–3.51]a	29 (100%)	2.87 [1.67–3.40]a	17 (77.27%)	0.92 [0.13–3.27]a

# Different bold letters in the same row indicate statistically significant differences between the groups (Wilcoxon rank-sum test with continuity correction). Nd, not detected in any sample of the group. LLOQ: lower limit of quantification.

**Table 3 viruses-15-01404-t003:** Frequencies of detection and the concentrations of the selected immune factors in the fecal samples of the three study groups (YHA, ELD1, and ELD2) #. All of the concentrations are expressed as ng/L, with the exception of BAFF/TNFSF13B, APRIL/TNFSF13, and chitinase 3-like 1, which are expressed as μg/L.

	YHA		ELD2		ELD1	
	n (%)	Concentration	n (%)	Concentration	n (%)	Concentration
APRIL/TNFSF13	9 (45%)	374.76 [311.7–883.66]a	16 (55.17%)	567.47 [<77.22–1178.34]a	12 (54.55%)	681.695 [<77.22–1930.44]a
BAFF/TNFSF13B	20 (100%)	304.365 [184.53–371.25]a	29 (100%)	997.54 [745.03–1090.67]b	22 (100%)	1214.35 [958.46–1513.80]b
Chitinase 3-like 1	20 (100%)	298.02 [223.71–462.86]a	28 (96.55%)	367.09 [314.78–467.11]a	21 (95.45%)	337.58 [206.45–538.33]a
gp130/sIL-6Rb	20 (100%)	64.31 [57.69–96.20]a	29 (100%)	69.69 [63.39–75.41]a	20 (90.91%)	69.47 [58.25–91.12]a
IL8	7 (35%)	0.31 [0.11–0.51]a	15 (51.72%)	0.71 [0.31–3.45]a	8 (36.36%)	0.51 [0.29–2.19]a
IL19	14 (70%)	3.735 [<0.95–4.67]a	9 (31.03%)	0.95 [<0.95–2.48]b	7 (31.82%)	0.95 [<0.95–1.11]b
IL26	11 (55%)	0.24 [<0.21–0.75]a	13 (44.83%)	<0.21 [<0.21–0.84]a	11 (50%)	0.32 [<0.21–0.72]a
IL32	18 (90%)	3.01 [2.13–4.53]a	29 (100%)	6.27 [4.92–6.96]b	21 (95.45%)	6.50 [5.68–7.83]b
IL34	15 (75%)	9.92 [4.42–21.24]a	23 (79.31%)	21.48 [11.45–30.24]a	16 (72.73%)	30.81 [4.24–41.46]a
IL35	15 (75%)	6.60 [1.91–13.19]a	21 (72.41%)	3.82 [<0.43–8.61]a	16 (72.73%)	4.44 [<0.43–9.76]a
LIGHT/TNFSF14	14 (70%)	0.35 [<0.09–0.72]a	17 (58.62%)	0.69 [<0.09–1.67]a	12 (54.55%)	0.29 [<0.09–1.30]a
Osteopontin	20 (100%)	80.84 [72.55–93.79]a	29 (100%)	81.72 [61.04–96.75]a	22 (100%)	88.62 [75.96–100.07]a
Pentraxin-3	20 (100%)	3.77 [2.98–4.06]a	18 (62.07%)	0.61 [<0.13–2.51]b	12 (54.55%)	0.215 [<0.13–0.58]b
sTNF-R1	14 (70%)	3.04 [<0.70–5.80]a	29 (100%)	24.76 [13.07–35.14]b	22 (100%)	29.62 [19.33–36.86]b
sTNF-R2	13 (65%)	0.69 [<0.12–1.64]a	29 (100%)	17.23 [15.23–21.56]b	22 (100%)	20.73 [16.29–26.30]b
TSLP	15 (75%)	0.71 [0.17–1.10]a	28 (96.55%)	1.58 [0.75–1.87]b	19 (86.36%)	0.94 [0.53–1.91]b
TWEAK/TNFSF12	20 (100%)	3.60 [2.68–5.04]a	29 (100%)	2.82 [2.02–4.01]a	22 (100%)	3.03 [2.31–4.22]a

# Different bold letters in the same row indicate statistically significant differences between the groups (Wilcoxon rank-sum test with continuity correction).

## Data Availability

Data supporting the reported results can be found in Appendix A.

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
