# Peer review of "Influence of SARS-CoV-2 Status and Aging on the Nasal and Fecal Immunological Profiles of Elderly Individuals Living in Nursing Homes"

_viruses, 2023, doi:10.3390/v15061404_

Round 1

Reviewer 1 Report

The research design in this study follows standard statistical research protocols and appears reasonably well done, and the reported findings are interesting. Here are a couple of things to attend to in a revision.

First, the presentation method chosen to indicate statistically significant differences between the groups needs to be improved. The font in which the contents of the tables are presented does not contain any bold letters. It contains only letters. This needs to be redone in such a way that the statistical differences are more clearly indicated, including which groups are being compared for the differences that are statistically significant.

Second, it would be good to indicate in the Discussion section how follow-up studies could be done that extend the findings reported in this paper, e.g., with period nasal and fecal samples from participants in longitudinal life course studies so that the adult life course trajectories of the immune factors could be studied and parameterized mathematically. 

Third, the paper could benefit from a careful edit of the English grammar. In most cases, this involves only minor wording edits. In others, you will need to make sure that your intended meanings are conveyed.

the paper needs a careful edit of the English grammar.  

As indicated to the authors, the paper needs a careful edit of the grammar.  

Author Response

First, the presentation method chosen to indicate statistically significant differences between the groups needs to be improved. The font in which the contents of the tables are presented does not contain any bold letters. It contains only letters. This needs to be redone in such a way that the statistical differences are more clearly indicated, including which groups are being compared for the differences that are statistically significant.

We agree that the presentation method chosen to indicate statistically significant differences between the groups needs to be improved. In the revised tables, we have added bold letters in the two tables to highlight those immune factors for which differences were detected among, at least, two groups.

Second, it would be good to indicate in the Discussion section how follow-up studies could be done that extend the findings reported in this paper, e.g., with period nasal and fecal samples from participants in longitudinal life course studies so that the adult life course trajectories of the immune factors could be studied and parameterized mathematically.

We thank the reviewer for this comment since it is a very relevant observation. In order to cope which this comment, we have added a sentence in the Discussion of the revised manuscript (lines 323 to 326), as follows:

In order to confirm or extend the findings reported in this work, nasal and fecal samples should be collected periodically from participants in longitudinal life course studies so that the adult life course trajectories of the immune factors could be studied and parameterized mathematically”.

Third, the paper could benefit from a careful edit of the English grammar. In most cases, this involves only minor wording edits. In others, you will need to make sure that your intended meanings are conveyed.

The revised manuscript has been revised by two native English speakers who have performed minor wording editing.

Reviewer 2 Report

1.  These authors have studied the immunological profiles in nasal secretions and fecal specimens in patients living in nursing homes which had COVID-19 infections or had no COVID-19 infections.  These patients were compared to healthy younger adults.  The authors measured a large number of immune factors in both nasal secretions and fecal specimens and made comparisons between the 2 elderly patient groups and the young healthy adults.
2.  It would be helpful if they provided slightly more information about the collection of nasal samples.  Can you consistently collect similar samples from multiple individuals?
3.  It would be helpful if the authors added bold numbers to table 2 to indicate to statistically significant differences, especially increased levels, in the group ELD1.
3.  The initial paragraphs in the results section on fecal samples can be somewhat confusing.  They report which immune factors were above the lower levels of quantification and the percentages for various factors above these levels.  It is not clear that this is very helpful.  It might be useful to provide some information on the correlation between nasals immune factor levels and fecal immune factor levels.
4.  In the discussion they note that the increased levels of IL-6, Il- 10 and tumor necrosis factor alpha in patients with SARS-CoV-2 infection probably represents an appropriate immune response.  They then comment that the higher levels in patients without SARS infection potentially represent an inappropriate inflammatory state based on immunoaging.  It is not clear to this reviewer that that is necessarily true.  Are there any studies which have measured the presence of these factors in patients at baseline and then determine the changes in these factors with infection?  Also is it known how stable the levels of these factors are over time?

Author Response

It would be helpful if they provided slightly more information about the collection of nasal samples. Can you consistently collect similar samples from multiple individuals?

We used the procedure described by Stewart et al. (2017) to collect the nasal samples because it is a standardized protocol that has been used in several cohort studies previously (Mansbach et al., 2012; Hasegawa et al., 2015) and it has been observed that allows proper analysis of proteins and metabolites present in the nasopharynx and has an excellent correlation when raw data and the rates between the concentration of the different soluble immune factors and the total protein concentration are compared. In order to cope with the reviewer’s comment, we have provided more information about the collection of the nasal samples in the revised manuscript (lines 70 to 76), as follows:

Briefly, 1 ml of normal saline was instilled into one nostril, and the mix of saline and mucus was collected using an 8F suction catheter. Then, the process was repeated in the other nostril. After sampling from both nares, the catheter was washed by suctioning 2 ml of saline, and the washing saline was added to the samples to ensure that a standard volume of aspirate was obtained. The samples were immediately placed on ice and then stored at −80°C until immunoprofiling.

We prefer the reference Stewart et al. (2017) rather than previous references using the same procedure (Mansbach et al., 2012; Hasegawa et al., 2015) because the description of Stewart et al. (2017) is more detailed than in the other papers.

It would be helpful if the authors added bold numbers to table 2 to indicate to statistically significant differences, especially increased levels, in the group ELD1.

In the revised tables, we have added bold letters to highlight those immune factors for which differences were detected among, at least, two groups.

The initial paragraphs in the results section on fecal samples can be somewhat confusing.  They report which immune factors were above the lower levels of quantification and the percentages for various factors above these levels.  It is not clear that this is very helpful.  It might be useful to provide some information on the correlation between nasals immune factor levels and fecal immune factor levels.

We agree that the initial paragraph in the results section on fecal samples can be somewhat confusing. We have modified it slightly in the revised manuscript (by deleting the reference to the lower levels of quantification). In addition, a new paragraph with a comparison of nasal and fecal values among the three study groups has been included in the revised manuscript, as follows (lines 203-209):

When the nasal and fecal samples were compared, some differences were found. All three groups (ELD1, ELD2, YHA) were significantly different from each other in relation to the nasal concentration of BAFF.TNFSF13B; in contrast, no differences were found in the fecal samples between the two elderly groups. In addition, the concentrations of some factors (APRIL.TNFSF13, LIGHT.TNFSF14, and osteopontin) showed to be statistically similar in the fecal samples of the three groups while their nasal concentrations were higher in the samples of the ELD1 group than in those from the YHA group.”.

A sentence related to differences in nasal and fecal profiles has also been included in the Discussion section (lines 289-294), as follows:

The fact that differences were found when some parameters were measured in different biological samples (nasal or fecal samples) provided by a same subject or by a same study group indicates that some cytokines and chemokines may have different patterns in different mucosal surfaces, even within a same host, and, therefore, confirms that the type of sample has to be taken into account when interpreting immunological data.

In the discussion they note that the increased levels of IL-6, Il- 10 and tumor necrosis factor alpha in patients with SARS-CoV-2 infection probably represents an appropriate immune response.  They then comment that the higher levels in patients without SARS infection potentially represent an inappropriate inflammatory state based on immunoaging.  It is not clear to this reviewer that that is necessarily true.  Are there any studies which have measured the presence of these factors in patients at baseline and then determine the changes in these factors with infection?  Also is it known how stable the levels of these factors are over time?

We thank the reviewer for this comment since it is a very relevant observation since there is a lack of longitudinal studies searching for the impact of different factors, including age and infections, on immunological parameters. In order to cope which this comment, we have added a sentence in the Discussion of the revised manuscript (lines 323 to 326), as follows:

In order to confirm or extend the findings reported in this work, nasal and fecal samples should be collected periodically from participants in longitudinal life course studies so that the adult life trajectories of the immune factors could be studied and parameterized mathematically”.

Reviewer 3 Report

The article entitled as "Nasal and fecal immunological profiling of elderly living in 2 SARS-CoV-2-positive and SARS-CoV-2-negative nursing 3 homes during the COVID-19 pandemia" is an interesting piece of research work. The article is written in a good way. However, consider the following suggestions to improve the scientific soundness and interest to the readers.

The Title needs to reframe, as it is giving an impression of like experiment or data analysis. Give it a interesting and catchy frame.

Additionally, my major concern is poorly written introduction, please update the introduction which will provide the significance of your study.   

In the discussion, Line no. 243 to 246: Authors have mentioned the importance of Th1 mediated cytokines.

It can also be mentioned that the existence of a Th1 cytokine profile has been linked to the severity of the infection, as a prolonged Th1 cytokine profile has been seen in severely infected patients with COVID-19 [Ref. Dhawan, M.; Rabaan, A.A.; Fawarah, M.M.A.; Almuthree, S.A.; Alsubki, R.A.; Alfaraj, A.H.; Mashraqi, M.M.; Alshamrani, S.A.; Abduljabbar, W.A.; Alwashmi, A.S.S.; Ibrahim, F.A.; Alsaleh, A.A.; Khamis, F.; Alsalman, J.; Sharma, M.; Emran, T.B. Updated Insights into the T Cell-Mediated Immune Response against SARS-CoV-2: A Step towards Efficient and Reliable Vaccines. Vaccines 202311, 101. https://doi.org/10.3390/vaccines11010101]. 

Like wise the discussion can be improved and updated and can be make more strong with respect to the results.

Further, as a reader I would like to read the conclusion as a separate section just to find the highlights of the findings, it will lead to the more reach of the manuscript. 

Best Wishes 

The English is readable and acceptable. However, need a second look to avoid any potential mistakes. 

Author Response

The Title needs to reframe, as it is giving an impression of like experiment or data analysis. Give it a interesting and catchy frame.

The title has been changed in order to cope with the reviewer’s suggestion. The new title is: “Influence of SARS-CoV-2 status and aging on the nasal and fecal immunological profiles of elderly living in nursing homes”.

Additionally, my major concern is poorly written introduction, please update the introduction which will provide the significance of your study.

The introduction has been modified in the revised manuscript, as follows (p. X-X):

In addition, immunosenescence and the so-called ”inflammaging” may play key roles in the enhanced severity of COVID-19 among elderly [10]. Such processes involve aging-related changes in both innate and adaptive immunity [11,12], which are associated to an increase in all-cause mortality. Impairment in triggering effective adaptive immune responses together with a basal pro-inflammatory state [13] may notably decrease the ability of elderly to a proper control of viral replication [14], predisposing them to more aggressive clinical consequences derived from cytokine storm and endothelial injury [10]. T cell responses are particularly affected, providing a rationale for the higher severity of COVID-19 among elderly. T cells, and particularly Th1 cells, play a key role in the pathogenesis and outcomes of SARS-CoV-2 infection and are closely related to inflammation [15]. In fact, a prolonged Th1 cytokine profile has been observed in severely infected COVID-19 patients [16, 17]. Most studies assessing the immune responses against SARS-CoV-2 have been focused in blood or bronchopulmonary samples; in contrast, studies investigating such responses in the other mucosal surfaces where the virus-host interactions start, such as that of the nasopharynx, are more scarce.

As a result, three new references have been included in the revised manuscript:

  1. Chauss, D., Freiwald, T., McGregor, R., Yan, B., Wang, L., Nova-Lamperti, E., Kumar, D., Zhang, Z., Teague, H., West, E. E., Vannella, K. M., Ramos-Benitez, M. J., Bibby, J., Kelly, A., Malik, A., Freeman, A. F., Schwartz, D. M., Portilla, D., Chertow, D. S., John, S., … Afzali, B. (2022). Autocrine vitamin D signaling switches off pro-inflammatory programs of TH1 cells. Nature immunology23(1), 62–74. https://doi.org/10.1038/s41590-021-01080-3
  2. Lucas, C., Klein, J., Sundaram, M. E., Liu, F., Wong, P., Silva, J., Mao, T., Oh, J. E., Mohanty, S., Huang, J., Tokuyama, M., Lu, P., Venkataraman, A., Park, A., Israelow, B., Vogels, C. B. F., Muenker, M. C., Chang, C. H., Casanovas-Massana, A., Moore, A. J., … Iwasaki, A. Nat Med. 2021;27(7):1178-1186. doi:10.1038/s41591-021-01355-0
  3. Dhawan, M., Rabaan, A. A., Fawarah, M. M. A., Almuthree, S. A., Alsubki, R. A., Alfaraj, A. H., Mashraqi, M. M., Alshamrani, S. A., Abduljabbar, W. A., Alwashmi, A. S. S., Ibrahim, F. A., Alsaleh, A. A., Khamis, F., Alsalman, J., Sharma, M., & Emran, T. B.  et al. Updated Insights into the T Cell-Mediated Immune Response against SARS-CoV-2: A Step towards Efficient and Reliable Vaccines. Vaccines (Basel). 2023;11(1):101. Published 2023 Jan 1. doi:10.3390/vaccines11010101

In the discussion, Line no. 243 to 246: Authors have mentioned the importance of Th1 mediated cytokines. It can also be mentioned that the existence of a Th1 cytokine profile has been linked to the severity of the infection, as a prolonged Th1 cytokine profile has been seen in severely infected patients with COVID-19 [Ref. Dhawan, M.; Rabaan, A.A.; Fawarah, M.M.A.; Almuthree, S.A.; Alsubki, R.A.; Alfaraj, A.H.; Mashraqi, M.M.; Alshamrani, S.A.; Abduljabbar, W.A.; Alwashmi, A.S.S.; Ibrahim, F.A.; Alsaleh, A.A.; Khamis, F.; Alsalman, J.; Sharma, M.; Emran, T.B. Updated Insights into the T Cell-Mediated Immune Response against SARS-CoV-2: A Step towards Efficient and Reliable Vaccines. Vaccines 2023, 11, 101. https://doi.org/10.3390/vaccines11010101]. Like wise the discussion can be improved and updated and can be make more strong with respect to the results.

Thank you for your comment. It has been taken into account in the revised manuscript, not only in the Discussion section but, also, in the Introduction section (see the answer to your previous comment).

Further, as a reader I would like to read the conclusion as a separate section just to find the highlights of the findings, it will lead to the more reach of the manuscript. 

We agree that it would be better to have a conclusion section. As a consequence, we have included such a section in the revised manuscript (lines 313-328), as follows:

“Nasal concentrations of BAFF/TNFSF13B, IL6, IL10 and TNF-α were significantly higher in samples from elderly than in those from younger healthy adults. Within elderly, they were higher among elderly living in a nursing home with a high rate of SARS-CoV-2-infected subjects, indicating that they can be considered as immunological hallmarks of SARS-CoV-2 infection at the nasopharyngeal level. Nasal and fecal concentrations of a wide number of pro-inflammatory cytokines were similar in the ELD1 and ELD2 groups but higher than those found in the YHA samples. These results reinforce the hypothesis that immunosenescence and inflammaging rendered elderly as a highly vulnerable population to a neo-infection, such as COVID-19 was during the first pandemic waves.

In this context, the increased nasal and fecal concentrations of most of the pro-inflammatory cytokines analyzed in this study in the ELD2 group, in relation to the YHA group, predicts that if the participants of these two groups were infected by SARS-CoV-2, the consequences for those in the ELD2 group would be much worse than among the YHA population. In order to confirm or extend the findings reported in this work, nasal and fecal samples should be collected periodically from participants in longitudinal life studies so that the adult life course trajectories of the immune factors could be studied and parameterized mathematically.”.

Round 2

Reviewer 1 Report

This revised paper has been improved and now is acceptable for publication.

This revised paper has been improved and now is acceptable for publication.